# Prospective Validation of the ROL System in Substaging pT1 High-Grade Urothelial Carcinoma: Results from a Mono-Institutional Confirmatory Analysis in BCG Treated Patients

**DOI:** 10.3390/cancers15030934

**Published:** 2023-02-01

**Authors:** Marina Valeri, Roberto Contieri, Vittorio Fasulo, Martina Iuzzolino, Miriam Cieri, Grazia M. Elefante, Camilla De Carlo, Alessandra Bressan, Cesare Saitta, Andrea Gobbo, Pier Paolo Avolio, Valerio Dacrema, Massimo Lazzeri, Gianluigi Taverna, Luigi M. Terracciano, Rodolfo Hurle, Piergiuseppe Colombo

**Affiliations:** 1Department of Pathology, IRCCS Humanitas Research Hospital, Via Manzoni 56, Rozzano, 20089 Milan, Italy; 2Department of Biomedical Science, Humanitas University, Via Rita Levi Montalcini 4, Pieve Emanuele, 20090 Milan, Italy; 3Department of Urology, IRCCS Humanitas Research Hospital, Via Manzoni 56, Rozzano, 20089 Milan, Italy; 4Department of Pharmacy, IRCCS Humanitas Research Hospital, Via Manzoni 56, Rozzano, 20089 Milan, Italy; 5Department of Urology, Humanitas Mater Domini, Via Gerenzano 2, Castellanza, 21053 Varese, Italy

**Keywords:** BCG, bladder cancer, non-muscle-invasive bladder cancer, prospective validation, pT1 high-grade urothelial carcinoma, risk stratification, ROL, substaging, TURBT, urothelial carcinoma

## Abstract

**Simple Summary:**

The management of patients with non-muscle-invasive, high-grade urothelial carcinoma represents a challenging issue for urologists. The ROL system is a method to evaluate tumor invasion and substage pT1 urothelial carcinoma. In this study, we aimed to confirm in a large and prospective series of cases that the ROL system significantly predicts tumor progression. We suggest the application of this system to improve clinical decision-making since it is easy to use, reproducible, and correlates not only with progression but also with recurrence.

**Abstract:**

Patients with pT1 high-grade (HG) urothelial carcinoma (UC) and a very high risk of progression might benefit from immediate radical cystectomy (RC), but this option remains controversial. Validation of a standardized method to evaluate the extent of lamina propria (LP) invasion (with recognized prognostic value) in transurethral resection (TURBT) specimens is still needed. The Rete Oncologica Lombarda (ROL) system showed a high predictive value for progression after TURBT in recent retrospective studies. The ROL system was supposed to be validated on a large prospective series of primary urothelial carcinomas from a single institution. From 2016 to 2020, we adopted ROL for all patients with pT1 HG UC on TURBT. We employed a 1.0-mm threshold to stratify tumors in ROL1 and ROL2. A total of 222 pT1 HG UC were analyzed. The median age was 74 years, with a predominance of men (73.8%). ROL was feasible in all cases: 91 cases were ROL1 (41%), and 131 were ROL2 (59%). At a median follow-up of 26.9 months (IQR 13.8–40.6), we registered 81 recurrences and 40 progressions. ROL was a significant predictor of tumor progression in both univariable (HR 3.53; CI 95% 1.56–7.99; *p* < 0.01) and multivariable (HR 2.88; CI 95% 1.24–6.66; *p* = 0.01) Cox regression analyses. At Kaplan-Meier estimates, ROL showed a correlation with both PFS (*p* = 0.0012) and RFS (*p* = 0.0167). Our results confirmed the strong predictive value of ROL for progression in a large prospective series. We encourage the application of ROL for reporting the extent of LP invasion, substaging T1 HG UC, and improving risk tables for urological decision-making.

## 1. Introduction

The management of patients with high-grade (HG) non-muscle invasive bladder cancer (NMIBC) remains a challenging issue in urological practice [1,2]. In particular, it is still debated when it is appropriate to perform an immediate radical cystectomy (RC) in this subgroup of patients. Indeed, RC could be an effective treatment in selected pT1 HG cases, while it might represent a potential overtreatment for others. Over the past decades, many efforts have been made to improve risk stratification and identify those patients who may benefit from immediate radical treatment; some features have demonstrated a solid predictive role and are currently employed [3,4,5,6,7,8,9,10,11,12,13,14].

Recently, the European Association of Urology (EAU) guidelines introduced an updated risk group system based on disease progression risk of [2,15]. The updated system introduced the group of very high-risk (VHR) patients in addition to the three already existing low, intermediate, and high-risk groups. Guidelines suggest discussing immediate RC with VHR patients [16].

Although not yet included in any guideline, a consensus was reached on the prognostic value of assessing the extent of lamina propria (LP) invasion in transurethral resection of bladder tumor (TURBT) specimens. Therefore, the newest World Health Organization (WHO) Classification of the Urinary and Male Genital Tumors strongly encourages pathologists to report this feature [17]. Despite this recommendation, validation of a gold standard method able to produce reliable pT1 substaging is still needed. Indeed, over the last years, different pT1 substaging approaches have been proposed [14,15,16,17,18,19,20,21]. To date, the anatomy-based method is one of the most widely applied, employing the histological landmark of the *muscularis mucosae* layer to produce a three- or a simplified two-tiered system (T1a/b/c or T1a/b) [18]. On the other hand, the size-based approaches adopt micrometric measurements of the maximum extent of LP invasion in any direction [21,22,23,24,25]. These systems showed clinical significance and overcame the challenging evaluation of the *muscularis mucosae* layer in TURBT specimens due to lack of orientation, possible hyperplastic appearance, anatomical variations, or total absence [20]. However, the most effective size-based method has not yet been identified [26,27].

Our approach is called the Rete Oncologica Lombarda (ROL) system; it has been developed over the last decade thanks to the collaboration of three large institutions in Northern Italy. ROL is a size-based system employing a simple 1.0 mm threshold, corresponding approximately to the diameter of a 20-power field (objective 20×). We have recently demonstrated that ROL is more feasible compared to other substaging methods and has a high predictive value for tumor progression after TURBT [28,29]. Nevertheless, our previous analyses were limited by their retrospective nature. Thus, in this work, we present the results of a prospective study aiming to validate the ROL system’s predictive value on a large, mono-institutional series of primary pT1 high-grade (HG) urothelial carcinomas (UC) treated with intravesical Bacillus Calmette-Guerin (BCG).

## 2. Materials and Methods

### 2.1. Patients

From January 2016 to December 2020, we prospectively maintained a database of all patients with a first diagnosis of pT1 HG UC who were then treated with BCG in a tertiary research hospital. Cases with a histotype different from transitional muscle invasion (pT2) at re-staging TUR (reTURBT), incorrect grading, or incomplete follow-up data were excluded. The maintenance scheme with BCG and follow-up were provided in accordance with the updated European guidelines (16). All patients completed the BCG induction course. Detailed clinicopathologic data were registered in the database.

### 2.2. Pathological Evaluation and Substage Attribution

Using the ROL system to assess LP invasion and to substage pT1, we adopted a cut-off of 1 mm (corresponding to the diameter of a high-power field, HPF, objective 20×, ocular 10×/field 22, diameter 1 × 1 mm) on hematoxylin and eosin slides. Tumors were stratified in ROL1 and ROL2 (Figure 1). ROL1 was defined as follows: (1) a single focus of LP invasion extending for ≤1.0 mm or (2) multiple foci of LP invasion extending for ≤1.0 mm summed together. ROL2 presented: (1) a single focus of LP invasion extending >1.0 mm or (2) multiple foci of LP invasion extending >1.0 mm summed together. The number of slides ranged from one to 14, depending on the size of the resected tumor. All slides were reviewed independently by three expert uropathologists to assign substaging and record pathologic features. Cases with discordant results were collectively discussed to reach a consensus.

### 2.3. Statistical Analysis

The endpoint was to assess and confirm the predictive value of ROL in terms of progression to muscle-invasive bladder cancer (MIBC) and recurrence-free survival after TURBT. Progression was defined as the diagnosis of a MIBC or of a distant metastasis either at TURBT or RC. Recurrence was defined as a relapsing pT1 or lower-stage tumor.

Time to event was calculated as the number of months between TUR and the event. Patients who did not recur or progress were censored at the date of death or the last follow-up visit. The characteristics of the patients were reported as descriptive statistics. Pearson’s chi-squared test and Wilcoxon’s rank sum test were used to compare categorical and continuous variables, respectively. Univariable and multivariable Cox regression analyses were used to identify significant independent predictors of progression after TUR. Kaplan-Meier (KM) survival estimates were used to investigate ROL’s correlation with progression-free survival (PFS) and recurrence-free survival (RFS). A two-sided *p*-value (*p*) < 0.05 was considered statistically significant. Analyses were performed with STATA 17.0 (StataCorp, College Station, TX, USA).

## 3. Results

A total of 284 patients with a new diagnosis of pT1 HG UC between January 2016 and December 2020 entered the prospective study. Of these, ten patients re-staged as pT2 at the second TUR were initially excluded. At slide review, 25 cases were excluded for incorrect grading (n = 12), absence of clear LP infiltration (n = 4), and histotype other than transitional (n = 9). Eventually, 13 patients were excluded due to incomplete follow-up data and another 14 due to incomplete BCG induction. As a result, a total of 222 patients with confirmed urothelial pT1 HG UC and complete follow-up data were analyzed. Clinico-pathologic features of the patients are summarized in Table 1. The median age was 74 years (interquartile range (IQR): 67–80), and most patients were males (73.8%). Sixty-nine patients presented with multifocal tumors (31.7%), and 33 cases had divergent differentiation (15%). Concomitant carcinoma in situ (CIS) and lymphovascular invasion (LVI) occurred in 31 (13%) and 18 (8.1%) cases, respectively.

The ROL system was feasible in all cases; 91 tumors were classified as ROL1 (41%), while 131 were substaged as ROL2 (59%). LVI, necrosis, and the presence of multiple foci of LP invasion were more present in ROL2 patients. Table 2 shows patients’ characteristics stratified according to ROL status. Representative cases are depicted in Figure 2.

At a median follow-up of 26.9 months (IQR 13.8–40.6), 81 patients recurred, while 40 patients progressed to MIBC. The 1-yr PFS rates were 93% (95% CI: 84.9–96.7) and 77% (95% CI: 69.0–83.9) for ROL1 and ROL2, respectively, while the 3-yr PFS rates were 92% (95% CI: 83.1–95.9) for ROL1 and 72% (95% CI: 62.0–79.6) for ROL2 (*p* = 0.0012). As for recurrence, the 1-yr RFS and 3-yr RFS were 86% (95% CI: 76.8–92.3) and 73% (95% CI: 61.5–81.9) for ROL1, respectively, and 66% (95% CI: 56.5–73.4) and 52% (95% CI: 40.7–62.0) for ROL2. We found a significant statistical difference in terms of time to recurrence between ROL1 and ROL2 (*p* = 0.0167).

At univariate Cox regression analysis, ROL emerged as a significant predictor of tumor progression (HR 3.53; CI 95% 1.56–7.99; *p* < 0.01). Multivariate analysis, including variables that might impact prognosis, showed that ROL was an independent predictor of progression (HR 2.88; CI 95% 1.24–6.66; *p* = 0.01) (Table 3). At KM estimates for PFS (Figure 3), we prospectively confirmed a significant ROL correlation with progression (*p* = 0.0012) (Figure 3B). Furthermore, ROL reached significance for RFS (*p* = 0.0167) (Figure 3C).

## 4. Discussion

Over the last decades, some clinical, pathological, and molecular features have shown a reliable negative predictive role and are currently used in the risk stratification of patients with pT1 HG UC [3,4,5,6,7,8,9,10,11,12,13,14]. Among the investigated histological characteristics, the value of assessing LP invasion has been extensively debated. The main concerns regarded interobserver variability and diagnostic pitfalls in staging superficial urothelial carcinoma in TURBT specimens, such as poor orientation, tangential sectioning, thermic injury, iatrogenic changes, or deceptive features like involvement of von Brunn’s nests, brisk inflammation, and pseudo-invasion [30,31,32,33].

Eventually, the prognostic value of assessing the extent of LP invasion in TURBT specimens reached a consensus in the scientific community. In keeping with this achievement, the authors of the newest WHO classification strongly recommend conveying the extent of LP invasion in the pathology reports using any of the methods proposed in the literature over the last few years [13,14,15,16,17,18,19,20,21], since a gold standard method has yet to be validated [26,27]. One of the most applied approaches is the anatomy-based approach, which uses the *muscularis mucosae* layer as a landmark to produce a substaging system (T1a/b/c or T1a/b) [18]. Nevertheless, the identification of *muscularis mucosae* in TURBT specimens is often challenging. Although some clues have been provided in the definition of the muscle layers and LP of the urinary bladder, lack of orientation of the fragments, hyperplastic *muscularis mucosae* hardly distinguishable from *muscularis propria*, and anatomic variations limit the reproducibility of the anatomic-based method [20]. In contrast, micrometric systems measuring the maximum extent of LP invasion at any direction overcome the anatomic issues and proved to have clinical significance. Interestingly, a recent retrospective study conducted on 73 patients comparing 6 different substaging methods showed that reporting the extent and/or the number of invasive foci represented the most practical approach and was not conditioned by orientation or artifacts [34]. The one proposed by van Rhijn et al. applies a 0.5 mm cut-off to classify tumors in T1m (microinvasive) and T1e (extensively invasive) [21]. Although others applied the same approach as van Rhijn’s [22], different studied demonstrated that the micrometric systems more feasible for a standardized usewere those employing the cut-off of 1.0 mm (23–25). Recently, de Jong et al. showed that T1 substaging (T1m/e) was an independent predictor of high-grade recurrence-free survival and progression-free survival in 264 patients treated with intravesical BCG [35].

In this setting, the ROL system, developed by our group, is a very simple micrometric approach, based on a 1.0 mm cut-off, with a more detailed and objective definition of the extent of LP invasion assessment, favoring reproducibility. Based on our experience, we believed that the daily practice might benefit from the possibility of adopting a 20× HPF diameter as a simplified threshold. In our large retrospective series of 314 patients with pT1 HG UC after TURBT, the impact of ROL on survival was compared to the anatomy-based approach (T1a/b) and the van Rhijn method (T1m/e). ROL and T1m/e were feasible in all cases, in contrast to T1a/b with only 72.3%, mainly due to the difficult identification of *muscularis mucosae* in the specimen. The ROL system alone correlated with PFS, while none of them predicted RFS [28]. Remarkably, in 2018, we reported similar results for a multi-institutional retrospective series of 250 transitional pT1 HG UC, with the ROL and van Rhijn systems being applicable in 99.6% of cases, whereas the feasibility of the anatomic approach was 76%. Consistently with the previous study, no system correlated with recurrence, and ROL was the only statistically significant predictor of progression [29]. These results were limited by the retrospective design of the studies. Therefore, we decided to conduct a prospective study aiming to confirm the predictive value of ROL for progression, adopting the ROL system from 2016 to 2020.

We here report the results of a prospective validation of the ROL system on a mono-institutional series of 222 primary pT1 HG urothelial carcinomas of the bladder treated with BCG [36]. ROL confirmed its high feasibility since it was applicable in 100% of cases. In retrospective studies, ROL was a significant predictor of progression in univariable analysis [28,29]. In this study, this evidence was supported for the first time by a reliable multivariate regression analysis (HR 2.88, *p* = 0.01). Importantly, ROL predicted progression independently and significantly, also when LVI was included in the analysis, which showed strong statistical significance in the univariable analysis for progression (HR 3.55, *p* < 0.01). At KM estimates, we prospectively confirmed that ROL significantly correlates with PFS (*p* = 0.0012). In addition, and in contrast with our previous findings, our results show a significant correlation with RFS (*p* = 0.0167). Our findings may benefit from the study’s prospective nature and, as a result, more accurate data registration and follow-up.

The study is not devoid of limitations. Firstly, the study involved a tertiary university hospital with a dedicated genitourinary pathology service. Consequently, it is impossible to draw any conclusions on the replicability of the ROL system on a daily basis among non-dedicated pathologists. Therefore, a multi-institutional prospective study is needed. Second, several T1 HG UC patients were treated with immediate RC and thus were not included in the study. This could represent a selection bias that may have excluded patients with very adverse outcomes. Furthermore, the decision to perform RC or bladder-sparing therapies following BCG failure was at the discretion of the clinician and, therefore, not standardized. Updated results after a longer FUP time may further confirm our findings. Moreover, it could be helpful to try to identify any differences in ROL application between “en bloc” and fragmented specimens, to date considered together, as well as any prognostic impact of separating ROL2 cases with small multiple foci of LP invasion extending for >1 mm from cases with a massive LP invasion, sometimes close to the *muscularis propria* [37]. From a technical point of view, although not applied in daily practice, it could be intriguing to analyze borderline cases (such as ROL1 with LP invasion close to 1 mm) through multiple sections at different levels in tumor blocks: the possible uncovering of a larger LP invasion might result in a substaging shift from ROL1 to ROL2. Additionally, ongoing studies are attempting to correlate pT1 substaging with the molecular subtypes included in the newest WHO classification, aiming to identify further predictors of progression and recurrence [38].

## 5. Conclusions

In conclusion, and in keeping with the suggestions of the newest WHO classification (17), we encourage the application of the ROL system for reporting the extent of LP invasion and for substaging pT1 HG UC. ROL is a simple and feasible method that might identify high-risk patients and eventually improve risk stratification and urological decision-making.

## Figures and Tables

**Figure 1 cancers-15-00934-f001:**
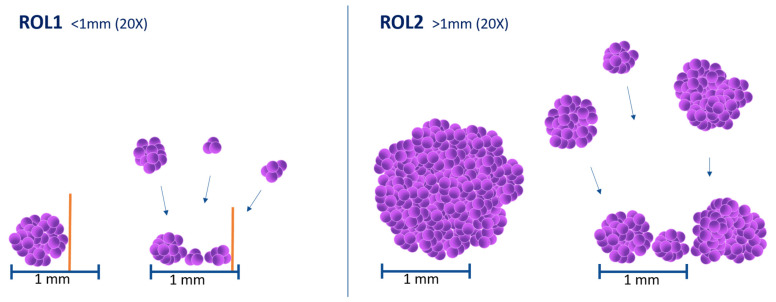
ROL system for substaging pT1 high-grade urothelial bladder carcinoma. ROL1 corresponded to a single or multiple foci of LP invasion extending for less than 1.0 mm at maximum extension and in any direction. ROL2 refers to a single or multiple foci of LP invasion measuring more than 1.0 mm. To simplify, the diameter of a 20×-power field (approximately 1.0 mm) can be employed. The sizes of the individual foci were summed to determine the ROL in cases of multifocal LP invasion. ROL: Rete Oncologica Lombarda; LP: lamina propria.

**Figure 2 cancers-15-00934-f002:**
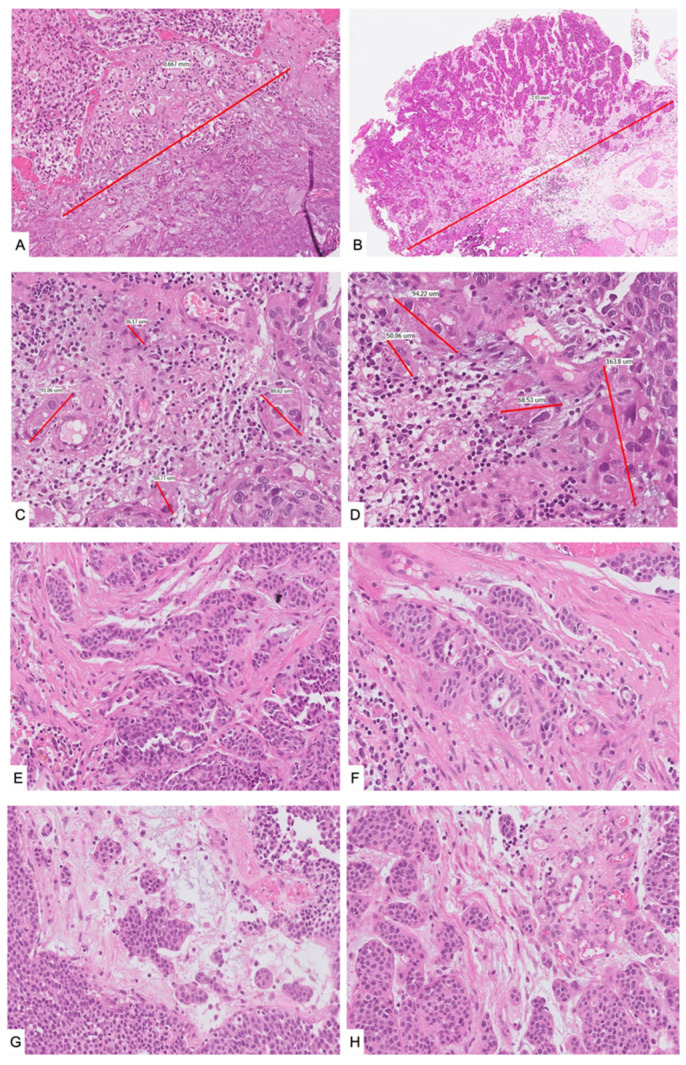
Examples of ROL substaging (Hematoxylin and Eosin). Single invasive foci measuring <1.0 mm (ROL1) ((**A**), 20×) and >1.0 mm (ROL2) ((**B**), 5×) at their greatest diameter in any direction. Multiple invasive foci of the same tumor extending together for <1.0 mm (ROL1) ((**C**,**D**), 20×) and for >1.0 mm (ROL2) ((**E**–**H**), 20×). ROL: Rete Oncologica Lombarda.

**Figure 3 cancers-15-00934-f003:**
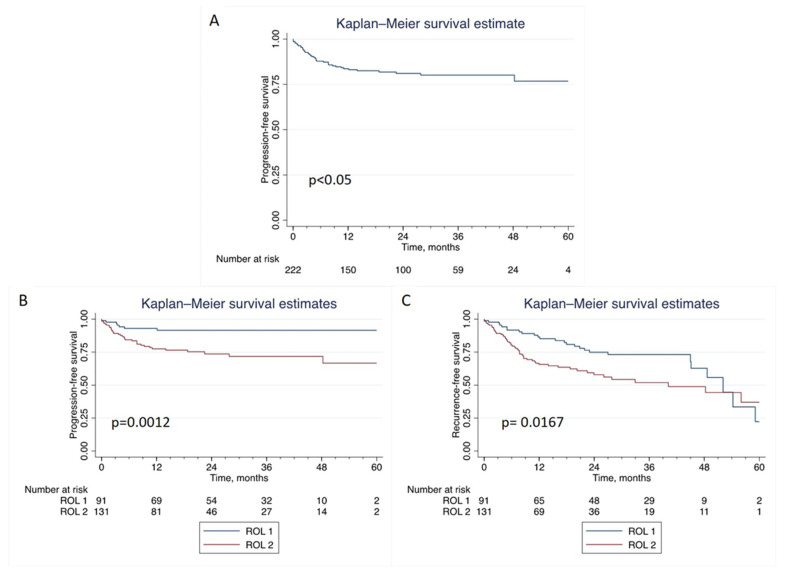
Kaplan-Meier (K-M) analyses for progression-free survival. (**A**) K-M estimates for progression-free (**B**) and recurrence-free (**C**) survival rates according to the ROL substaging system. ROL: Rete Oncologica Lombarda.

**Table 1 cancers-15-00934-t001:** Patients’ clinico-pathologic features.

Variable		Patients (n = 222)
**Age, y**	Median (IQR)	74 (67–80)
Gender, n (%)	Male	164 (73.8)
	Female	58 (26.2)
Smoking status	Never/Former	128 (57.6)
	Active	94 (42.4)
BMI	Median (IQR)	26.1 (23.4–28.7)
Multifocality, n (%)	No	153 (68.3)
	Yes	69 (31.7)
Tumor size, n (%)	<3 cm	152 (68.3)
	>3 cm	70 (31.7)
Histology, n (%)	Pure transitional	189 (85)
	Divergent differentiation	33 (15)
Associated CIS, n (%)	No	191 (86)
	Yes	31 (14)
LVI, n (%)	No	204 (91.9)
	Yes	18 (8.1)
Necrosis, n (%)	No	148 (66.7)
	Yes	74 (33.3)
LP invasion, n (%)	Single focus	77 (34.7)
	Multiple foci	145 (65.3)

IQR: interquartile range; CIS: carcinoma in situ; LVI: lymphovascular invasion; LP: lamina propria.

**Table 2 cancers-15-00934-t002:** Characteristics of patients stratified according to ROL status.

		ROL1 (*n* = 91)	ROL2 (*n* = 131)	*p*-Value *
Age, median (IQR)		75 (71–82)	73 (64.5–79)	0.08
Gender, n (%)	Male	66 (72.5)	98 (74.8)	0.70
	Female	25 (27.5)	33 (25.2)	
Smoking status	Never/Former	54 (59.3)	74 (56.5)	0.67
	Active	37 (40.7)	57 (43.5)	
BMI, median (IQR)		25.4 (22.6–28.7)	26.2 (23.7–28.5)	0.71
Recurrent tumor, n (%)	No	66 (72.6)	109 (83.2)	0.06
	Yes	25 (27.4)	22 (16.8)	
Multifocality, n (%)	No	63 (69.3)	90 (68.7)	0.98
	Yes	28 (30.7)	41 (31.3)	
Tumor size, n (%)	<3 cm	62 (68.1)	90 (68.7)	0.92
	>3 cm	29 (31.9)	41 (31.3)	
Concomitant CIS, n (%)	No	78 (85.7)	113 (86.3)	0.91
	Yes	13 (14.3)	18 (13.7)	
LVI	No	89 (97.8)	115 (87.7)	<0.01
	Yes	2 (2.2)	16 (12.3)	
Necrosis	No	69 (75.8)	79 (60.3)	0.01
	Yes	22 (24.2)	52 (39.7)	
Lamina Propria invasion	Single focus	59 (64.8)	18 (13.7)	<0.01
	Multiple foci	32 (35.2)	113 (86.3)	
Second resection	No	25 (27.5)	53 (40.4)	0.10
	Yes	66 (72.5)	78 (59.6)	
Recurrence	No	64 (70.3)	77 (58.8)	0.78
	Yes	27 (29.7)	54 (41.2)	

IQR: interquartile range; CIS: carcinoma in situ; LVI: lymphovascular invasion; LP: lamina propria.* Pearson’s chi-squared test was used to compare categorical variables; Wilcoxon’s rank sum test was used to compare continuous variables.

**Table 3 cancers-15-00934-t003:** Univariate and multivariate analysis of time to progression (40 events).

		Univariate		Multivariate		
Variable	HR	CI 95%	*p*-Value	HR	CI 95%	*p*-Value
ROL (1, 2)	3.53	1.56–7.99	<0.01	2.88	1.24–6.66	0.01
LVI (No, Yes)	3.55	1.69–7.46	<0.01	2.76	1.27–6.02	0.01
Number of tumors (Single, multiple)	1.53	0.80–2.92	0.19	1.71	0.89–3.30	0.11
Tumor dimension (<3 cm, >3 cm)	2.20	1.17–4.12	0.01	1.77	0.92–3.41	0.09
Concomitant CIS (No, Yes)	0.46	0.14–1.51	0.21	0.60	0.18–1.98	0.40
Necrosis (No, Yes)	1.93	1.04–3.60	0.04	1.32	0.68–2.57	0.42

HR: hazard ratio; CI: confidence interval; ROL: Rete Oncologica Lombarda; CIS: carcinoma in situ; LVI: lymphovascular invasion.

## Data Availability

The data presented in this study are available on request from the corresponding author. The data are not publicly available due to ethical considerations related to the privacy of the medical data of patients included in this study.

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
