# Peer review of "Prospective Validation of the ROL System in Substaging pT1 High-Grade Urothelial Carcinoma: Results from a Mono-Institutional Confirmatory Analysis in BCG Treated Patients"

_cancers, 2023, doi:10.3390/cancers15030934_

Round 1

Reviewer 1 Report

With great interest I read the manuscript of the authors, presenting a very fast forward approach of a pathological classification for pT1HG NMIBC, which is a challenging issue for Urologist. The manuscript is very well written and has a very clear message for the reader. I have some minor comments and Ideas to address:

Line 205 change Van Rhijn to van Rhijn

Do you have the data of BCG responders? Could be included in table 2

Table 3: Please include the number of events in the figure title (maybe in brackets), as there are 40 patients that experienced progression at 27 month FU, the multivariate model is at risk for overfitting (usually 10 events/variable), I would consider to remove age from the (multivariate) analysis

Author Response

We thank the reviewer 1 for the comments and suggestions. Here the responses point by point:

  1. Line 205 change Van Rhijn to van Rhijn

We thank the reviewer for the comment, we corrected (see line 215).

  1. Do you have the data of BCG responders? Could be included in table 2

We thank the reviewer for the comment. Yes, we have those data and we modified table 2 accordingly. As suggested by the reviewer, we reported the number of recurrences (yes, no) (corresponding to BCG failures and responders, respectively) stratified by ROL status.

  1. Table 3: Please include the number of events in the figure title (maybe in brackets), as there are 40 patients that experienced progression at 27 month FU, the multivariate model is at risk for overfitting (usually 10 events/variable), I would consider to remove age from the (multivariate) analysis

We thank the reviewer for the suggestion. We decided to repeat the analysis excluding age (see Table 3) and we also specified the number of events in brackets in the table title.

Reviewer 2 Report

The authors have prepared a nicely written and designed study on substaging pT1 high-grade urothelial carcinoma. Only minor issues are suggested to address as listed below.

1. Based on the way the authors described bladder cancers in their cohort, all bladder cancers were pT1 high-grade papillary urothelial carcinoma. I would suggest author to replace "bladder cancer" in the title and text with such. 

2. I would suggest authors to modify Figure 1or choose better illustration to clarify their method on ROL1 and ROL2. The current version using dots may be misleading as the dots in Figure 1 make readers think they represent individual tumor nests. 

Author Response

We thank the reviewer 2 for the comments and suggestions. Here the responses point by point:

  1. Based on the way the authors described bladder cancers in their cohort, all bladder cancers were pT1 high-grade papillary urothelial carcinoma. I would suggest author to replace "bladder cancer" in the title and text with such. 

We thank the reviewer for the suggestion. Accordingly, we modified to “urothelial carcinoma” throughout the text when referring to our cohort. We used a generic term because the cohort was composed of both papillary and non-papillary carcinomas.

  1. I would suggest authors to modify Figure 1 or choose better illustration to clarify their method on ROL1 and ROL2. The current version using dots may be misleading as the dots in Figure 1 make readers think they represent individual tumor nests. 

We thank the reviewer for the comment. Figure 1 is supposed to be a simplified scheme to explain how we measure not only a single, but also multiple invasive foci of lamina propria tumor infiltration. Indeed, the dots represent single invasive foci or nests of tumor cells infiltrating lamina propria. In accordance to the reviewer’s suggestions, we modified Figure 1 to better represent clusters of tumor cells.

Reviewer 3 Report

The authors performed a validation study to show that ROL score is predictable to prognosis of patients with high grade NMIBC. The paper shows that ROL is associated with progression-free survival time and clinical characteristics of patients. Before publishing, this authors need to complete some information.

1. In Table 2, how are the p-values calculated? What kind of statistic test (for each characteristics) is performed?

2. The first row of Table 2 shows that in ROL2 group, median age is 73 and IQR is 74.5-79. How could the first quantile (25%) be lower than median value?

3.  Table 3 shows that ROL is associated with prognosis. But ROL could be a dependent or redundant predictor. The authors should test whether ROL is an independent predictor.

4. The authors should compare ROL with existing substaging systems.

Author Response

We thank the reviewer 3 for the comments and suggestions. As suggested, we revised and improved English language throughout the text.

Here the responses point by point:

  1. In Table 2, how are the p-values calculated? What kind of statistic test (for each characteristics) is performed?

We thank the reviewer for the comment. As explained in the methods (line 129-131), we used Pearson’s chi-squared test and Wilcoxon’s rank sum test to comparing categorical and continuous variables, respectively. To clarify, we added a sentence in the table description (see * in table 2)

  1. The first row of Table 2 shows that in ROL2 group, median age is 73 and IQR is 74.5-79. How could the first quantile (25%) be lower than median value?

We thank the reviewer for the comment. We apologize for the occured mistake, we amended (see table 2).

  1. Table 3 shows that ROL is associated with prognosis. But ROL could be a dependent or redundant predictor. The authors should test whether ROL is an independent predictor.

We thank the reviewer for the comment. Indeed, since ROL substaging might be influenced by other variables, we decided to verify in a multivariate analysis its prognostic value and we included in this analysis all variables that might impact on prognosis. ROL was confirmed as an independent predictor of progression at multivariate analysis (HR 2.88; CI 95% 1.24 – 6.66; p=0.01). To clarify, we modified the sentence in the Results (see line 176-179)

  1. The authors should compare ROL with existing substaging systems.

We thank the reviewer for the comment. Our group has already compared ROL system to previously adopted systems (pT1a/b anatomy based approach and pT1m/e micrometric approach) in recent retrospective studies (Patriarca C. Diagn Pathol. 2016 Jan, 20, 11:6. Colombo R., et al. Eur Urol Focus. 2018 Jan, 4(1), 87-93.). ROL impact on survival (progression) was compared to the other approaches: we demonstrated that only ROL system correlated with PFS and was also more easy to use and possibly more reproducible compared with previous methods. This comparison was not the objective and focus of the present study. Here we aimed to validate prospectively the impact on progression of a daily practice employment of ROL system in pT1 high-grade urothelial carcinomas. In accordance with the suggestion of the reviewer, a future prospective comparison with the anatomic approach could be interesting, in particular with the aim of improving AJCC staging system for bladder cancer.